# Learning to navigate by distilling visual information and natural language instructions

## Abstract

In this work, we focus on the problem of grounding language by training an agent to follow a set of natural language instructions and navigate to a target object in a 2D grid environment. The agent receives visual information through raw pixels and a natural language instruction telling what task needs to be achieved. Other than these two sources of information, our model does not have any prior information of both the visual and textual modalities and is end-to-end trainable. We develop an attention mechanism for multi-modal fusion of visual and textual modalities that allows the agent to learn to complete the navigation tasks and also achieve language grounding. Our experimental results show that our attention mechanism outperforms the existing multi-modal fusion mechanisms proposed in order to solve the above mentioned navigation task. We demonstrate through the visualization of attention weights that our model learns to correlate attributes of the object referred in the instruction with visual representations and also show that the learnt textual representations are semantically meaningful as they follow vector arithmetic and are also consistent enough to induce translation between instructions in different natural languages. We also show that our model generalizes effectively to unseen scenarios and exhibit *zero-shot* generalization capabilities. In order to simulate the above described challenges, we introduce a new 2D environment for an agent to jointly learn visual and textual modalities.

## 1 Introduction

Understanding of natural language instructions is an important aspect of an Artificial Intelligence (AI) system. In order to successfully accomplish tasks specified by natural language instructions, an agent has to extract representations of language that are semantically meaningful and ground it in perceptual elements and actions in the environment.

Humans have the ability to understand the true essence of the words and thus they can easily decipher sentences even if it contains some new combination of words. It is not unreasonable to expect the same from an AI agent. The information extracted by agent from the language should be such that it corresponds to the true meaning of the word so that it enables the agent to generalize to even unseen combinations of words. For instance, when given sufficient information about the words such as 'green' and 'bag', it should automatically figure out as to what 'green bag' essentially means.

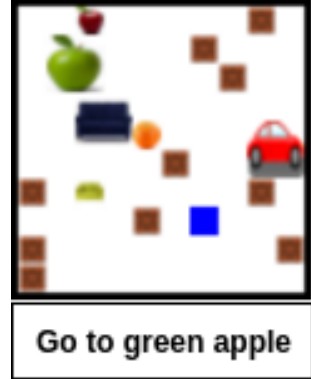

Figure 1: The agent (blue in color) should learn to read the instruction and navigate to green apple.

Consider a task in which an agent has to learn to navigate to a target object in a 2D grid environment as shown in figure 1. The environment consists of many objects with different attributes (in our case: green apple, red apple, blue sofa, green sofa, an orange fruit, red car) and multiple obstacles. The agent receives visual information through raw pixels and an instruction telling what task needs to be achieved. The challenges that the agent has to tackle here are manyfold: a) the agent has to develop the capability to recognize various objects, b) have some memory of objects seen in previous states

while exploring the environment as the objects may occlude each other and/or may not be present in the agent's field of view c) ground the instruction in visual elements and actions in the environment and d) learn a policy to navigate to the target object by avoiding the obstacles and other non-target objects.

We tackle this problem by proposing an end-to-end trainable architecture that creates a combined representation of the image observed by the agent and the instruction it receives. Our model does not have any prior information of both the visual and textual modalities. We develop an attention mechanism for multimodal fusion of visual and textual modalities. Our experimental results show that our attention mechanism outperforms the existing multimodal fusion mechanisms proposed in order to solve the above mentioned task. We demonstrate through the visualization of attention weights that our model learns to correlate attributes of the object referred in the instruction with visual representations and also show that the learnt textual representations are semantically meaningful as they follow vector arithmetic and are also consistent enough to induce translation between instructions in different natural languages. We also show that our model generalizes effectively to unseen scenarios and exhibit *zero-shot* (ZS) generalization capabilities. In order to simulate the above described challenges, we introduce a new 2D environment for an agent to jointly learn visual and textual modalities. Our 2D environment is also thread compatible. Finally, in order to enable the reproducibility of our research through participation in Pineau (2017) and foster further research in this direction, we open source our environment as well as the code and models that we developed.

## 2 RELATED WORK

The task of grounding natural language instructions has been well explored by researchers in different domains. From the robotics domain : (Guadarrama et al. (2013), Tellex et al. (2011)) focus on grounding verbs in navigational instructions like *go*, *pick up*, *move*, *follow* etc. (Chao et al. (2011), Lemaignan et al. (2012)) ground various concepts through human-robot interaction. (Artzi & Zettlemoyer (2013), Misra et al. (2016)) focus on grounding natural language instructions by mapping instructions to action sequences in 2D and 3D environments respectively. Given a natural language instruction, Chen & Mooney (2011) attempt to learn a navigation policy that is optimal in a 2D maze-like setting by relying on a semantic parser. Mei et al. (2016) focus on neural mapping of navigational instructions to action sequences by representing the state of the world using bag-of-words visual features.

Deep reinforcement learning agents have been previously used to solve tasks in both 2D Mnih et al. (2015) and 3D Mnih et al. (2016) environments. More recently, these approaches have been used for playing first-person shooter games in Lample & Chaplot (2017) and Kempka et al. (2016). These works focus on learning optimal policy for different tasks using only visual features, while our work involves the agent receiving natural language instruction in addition to visual state of the environment.

The authors in Yu et al. (2017) and concurrent ICLR 2018 submission Anonymous1 (2018) propose an end-to-end framework by which an agent learns to navigate in 2D maze-like environment (XWORLD) using natural language instructions. They simultaneously learn visual representations, syntax and semantics of natural language instruction as well as the navigation action to perform. The task of the agent is basically navigation plus Visual Question Answering (VQA) Antol et al. (2015); the agent at every step either gets a navigation instruction or a question about the environment, and the output is either a navigation action or answer to the question posed. Another concurrent ICLR 2018 submission Anonymous2 (2018) use a simple concatenation of visual and textual representations. Chaplot et al. (2017) propose a Gated-Attention architecture for task oriented language grounding and evaluate their approach on a new environment built over VizDoom Kempka et al. (2016). They do hadamard product between textual and visual representations, whereas we generate multiple textual embeddings by passing Gated Recurrent Unit (GRU) Chung et al. (2014) features to parallel Fully-Connected (FC) layers and then use each of them to convolve with the visual features thus generating multiple attention maps.

In our work, we present a simple attention based architecture in order to ground natural language instructions in a 2D grid environment. Our model does not have any prior information of both the visual and textual modalities and is end-to-end trainable without the requirement of external semantic parser or language models as with prior work in robotics domain mentioned previously. Through

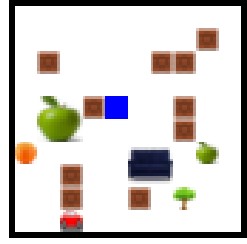 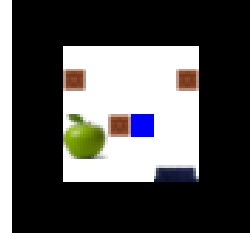

(a) Complete view of environment.      (b) Egocentric view as seen by agent.

Figure 2: Example state of environment.

our multimodal fusion mechanism, we obtain a joint concise representation of both the visual and textual modalities which is sufficient for the agent to learn optimal policy. When compared to prior work Yu et al. (2017) and concurrent ICLR 2018 submission Anonymous1 (2018), the scenarios we work with are more complex as our environment has a larger grid size, more number of objects present concurrently in it, increased complexity of natural language instructions (two sentence instructions). In addition, the new environment introduced by us is thread compatible.

## 3 PROBLEM DESCRIPTION

We tackle the problem in which an agent learns to navigate to the target object in a 2D grid environment. The agent receives a natural language instruction at the beginning of every episode which specifies the characteristics of the target object. The episode terminates when the agent reaches the target object or the number of time steps exceed the maximum episode length. The environment consists of several other objects as well as many obstacles which act as distractors. The agent receives a negative reward when it reaches any non-target object or an obstacle. The agent sees an egocentric view of the environment in which the objects present outside a certain predefined radius are not visible and hence the agent does not have complete knowledge about the environment. The objective of the agent is to learn an optimal policy so that it reaches the correct object before the episode terminates.

## 4 ENVIRONMENT

We create a new 2-D grid based environment in which the agent interacts with the environment and performs one of the four actions: *up*, *down*, *left* and *right*. Each scenario in the environment consists of an agent, a list of objects and a list of obstacles (walls) as shown in figure 2a. Every object and obstacle has a set of attributes associated with it like color, size etc. Environment is completely customizable as the grid size, number of objects, type of objects, number of obstacles along with their corresponding attributes can be specified in a configurable specification file which is in JSON format (refer section 9.6 for an example configuration file). The agent perceives the environment through raw RGB pixels with an egocentric view, in which we blacken out the region outside certain predefined radius centered at the agent(figure 2b). At the beginning of every episode, the agent receives a natural language instruction (Eg: "Go to red apple") and obtains a positive reward (+1) on successful completion of the task. The task is considered successful if the agent is able to navigate to the target object (red apple in this case) correctly before the episode ends (which is specified by *max_episode_length*). The agent gets a negative reward (-1) whenever it hits a wall. The agent also receives a small negative (-0.5) reward if it reaches any non-target object and the non-target object is then placed at a new random location within the grid. In every episode, the environment is reset randomly, i.e., the agent, the target object, non-target objects are placed at random locations in the grid. This means that the objects may occlude each other and/or may not be in the agent's egocentric view in the initial configuration. The instruction is generated based on the initial configuration of the environment. Some example instructions are given below:

- *Go to car* or *Car is your target(or destination)*. If the environment consists of multiple cars that differ in their size and/or color attributes, the agent gets a positive reward if it navigates to any one of them.

- *Go to green apple* or *Green apple is your target(or destination)*. The agent gets a positive reward only if it reaches the green apple. These types of instructions test the agent's capability to distinguish between objects on the basis of color.

- *Go to medium blue sofa* or *Medium blue sofa is your target(or destination)*. If there are multiple blue sofas in the environment which differ in their size attribute, the agent gets a positive reward only if it navigates to the blue sofa with medium size. These type of instructions demonstrate the agent's ability to distinguish between multiple instances of the same object which have different sizes based on visual appearance.

- *Go to south of blue bus*. The agent receives a positive reward only if it reaches the grid which is to the south of blue bus.

- *Go to bottom right corner*. The agent gets a positive reward only if it reaches the bottom-most and rightmost grid in the environment.

- *There is a orange chair. Go to it*. The environment can consist of multiple instances of orange chair which differ in their size attribute and the agent gets a positive reward if it reaches any one of them.

- *There is a small blue bag. Go to it*. The environment can consists of multiple instances of blue bag which differ in their size attribute, however the agent gets a positive reward only if it reaches the small blue bag. These type of instructions demonstrate the agent's ability to distinguish between multiple instances of the same object which have different sizes based on visual appearance.

- *There are multiple green tree. Go to smaller(or larger) one*. These type of instruction are generated only when there are multiple instances of an object, distinguished on the basis of size are present in the environment. The agent is required to correlate the word smaller(or larger) with the corresponding size of object and then navigate to it accordingly.

- *There is a small yellow banana and a medium black television. Go to former(or latter)*. The agent is required to understand the word former or latter and go to either small yellow banana or medium black television respectively.

- *If small yellow flower is present then go to small orange cat else go to medium black bear*. The agent is required to check if small yellow flower is present in the environment or not and accordingly navigate to the correct object.

The complete set of objects and instructions are provided in section 9.

## 5 PROPOSED APPROACH

In order to solve the above mentioned problem, we propose an architecture which can be trained end-to-end and does not have any prior knowledge of visual and textual modalities. The proposed model can be divided into three phases a) input processing phase b) multi-modal fusion phase and c) policy learning phase, as illustrated in figure 3

### 5.1 INPUT PROCESSING PHASE

The agent at each time step ($t$) receives a RGB image of the environment which is egocentric ($E_t$). It also receives an instruction ($I$) at the start of every episode, which contains the details of what task needs to be performed by the agent. The image $E_t$ is passed through several layers of convolutional neural network (CNN) to obtain a good representation of the image. We do not use pooling layers because pooling leads to translational invariance, however in case of our problem the location of the target object, non-target object and obstacles are crucial in determining the potential reward. The final representation of the image obtained in this phase is $R_E$ and is of dimension $W \times H \times D$, where $W$, $H$ are width and height of each feature map and $D$ represents the number of feature maps. In our case the image representation has 64 channels, i.e. $D = 64$ and $W = H = 7$ (Refer section 6.1 for exact details of network hyper-parameters). Every word in the instruction $I$ is converted into an one

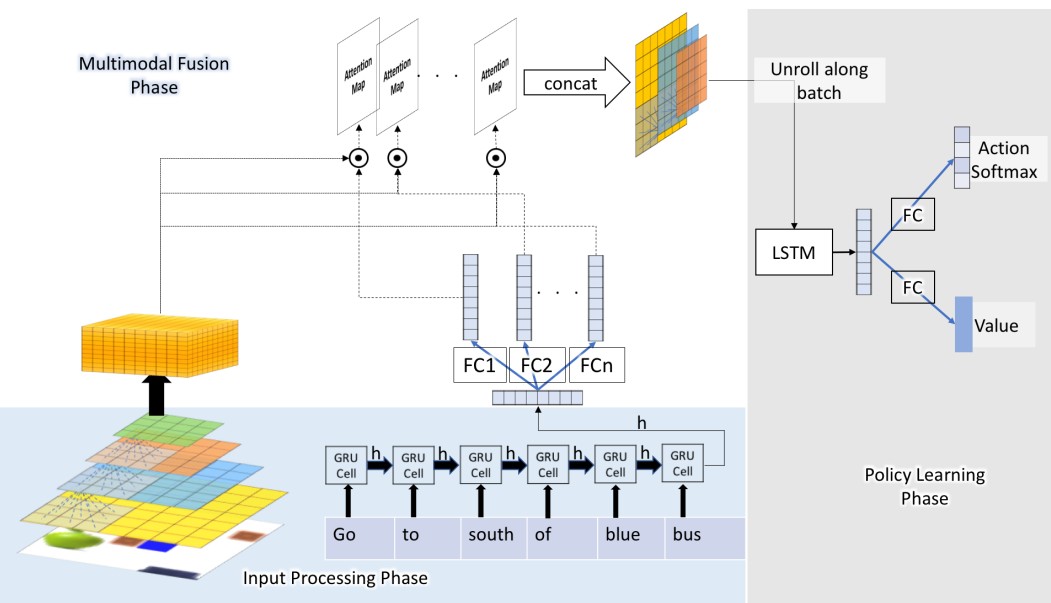

Figure 3: Our network architecture consisting of all three phases.

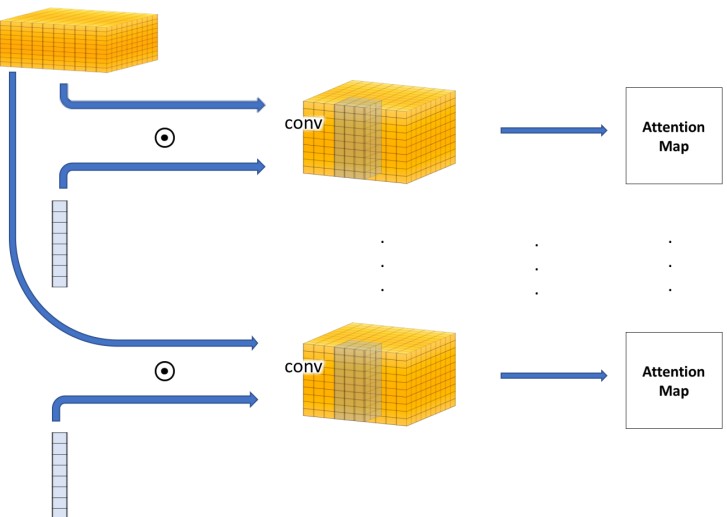

Figure 4: Multimodal fusion phase: each vector obtained from FC layer is used as 1x1 filter to perform convolution over visual representation.

hot vector and then concatenated before passing it through the GRU. In all our evaluation scenarios, the maximum number of words present in the instruction is 18. The total number of possible unique words (i.e. *vocabulary size*) is 72. The complete list of these words are provided in section 9.4. Sentences that have less than 18 words are zero padded to ensure that all sentences are represented by 18 one hot vectors. The concatenated one hot vectors are passed through a GRU and the output of the final time step is used as a representation of the entire sentence. The output of the GRU is then passed through multiple parallel (say $n$) FC layers each of size 64 (equal to the number of feature maps in the final image representation) and reshaped into $1 \times 1 \times 64 \times 1$. This results in generation of vectors $V_1, V_2 \ldots, V_n$

## 5.2 MULTIMODAL FUSION PHASE

We propose an attention method for the multimodal fusion of $R_E$ and $V_j$ where $j \in \{1,2, \ldots, n\}$. Each $V_j$ is used as a $1 \times 1$ filter to perform 2D convolution over feature volume $R_E$ which is of dimension $7 \times 7 \times 64$. This is feasible because both $R_E$ and $V_j$, each have 64 channels. Each such fusion results in an attention map of dimension $7 \times 7 \times 1$ where the value of each pixel is the dot product across channels of the filter with the corresponding pixel in the visual representation (figure 4 provides an illustration of this phase). All these attention maps are then concatenated to result in a tensor of dimension $7 \times 7 \times n$ (say $M_{attn}$ and the corresponding model $A3C_{attn}$). In our experiments we try with $n \in \{1,5,10\}$ (refer section 5 for comparison of performance). This attention mechanism draws inspiration from Xu et al. (2015). The visual representation obtained through multiple layers of convolutions can detect various attributes (such as color, shape, size) of the objects in the environment. The agent is required to attend to specific attributes of the objects based on the instruction. For example, when the instruction is "Go to blue sofa", the agent should learn to attend to objects that are 'blue' in color as well as attend to objects that resemble a 'sofa'. We demonstrate through the visualization of attention weights in section 5 that our attention method learns to correlate attributes of the object referred in the instruction with visual representations. $M_{attn}$ is then fed as input to the policy learning phase. Through this fusion mechanism, we have obtained a concise representation ($M_{attn}$) of both the textual and visual modalities which symbolizes consciousness of an AI agent as highlighted in Bengio (2017). We also consider a small variant of this fusion mechanism where we concatenate one of the attention maps with the visual representation to result in a tensor of dimension $7 \times 7 \times 65$ (say $M_{netattn}$ and the corresponding model $A3C_{netattn}$). We show in section 5 that the policy learnt by the agent when $M_{netattn}$ is fed as input to the policy learning phase performs poorly on our evaluation scenarios when compared to the case when $M_{attn}$ is fed as input to the policy learning phase. We also compare our method with previously proposed multimodal fusion methods: a) Gated Attention unit Chaplot et al. (2017) (say $A3C_{hadamard}$) b) Concatenation unit Yu et al. (2017) (say $A3C_{concat}$). We replicate the complete architecture along with the hyperparameters used by the authors in Chaplot et al. (2017) for our comparison.

## 5.3 POLICY LEARNING PHASE

This phase receives as input, the output of the multimodal fusion phase. We adopt a reinforcement learning approach using the Asynchronous Advantage Actor-Critic (A3C) Mnih et al. (2016) algorithm. In A3C, there is a global network and multiple local worker agents each having their own network parameters. Each of these worker agents interacts with it's own copy of environment simultaneously and gradients of these worker agents are used to update the global network. Refer section 6.1 for exact details of our architecture.

## 6 EXPERIMENTAL SETUP

In all our experiments, we fix the grid size to be 10x10 with 63 objects and 8 obstacles. At the start of every episode, we randomly select between 3 to 6 objects from the set of 63 objects and a single instruction is randomly selected from a set of feasible instructions which are generated based on the initial configuration of the environment. Since the agents, objects and obstacles are placed at random locations in the grid, we take into consideration the reachability of the agent to various objects while generating the set of feasible instructions for the given episode. The episode terminates if the agent navigates to the target object or when time steps exceed the maximum episode length. We use *accuracy* of the agent and *mean reward* it obtains as the evaluation metrics. The *accuracy*

is the number of times the agent reaches the target object successfully before the episode terminates and *mean reward* is the average reward that the agent obtains across all episodes. We consider two modes of evaluation:

1) **Unseen scenario generalization:** At test time, the agent is evaluated in unseen environment scenarios with instructions in the train set. The scenario consists of combination of objects placed at random locations not seen before by the agent at train time.

2) **Zero-shot generalization:** At test time, the agent is evaluated with unseen instructions that consists of new combinations of attribute-object pairs never seen during training. The environment scenario is also unseen in this case.

## 6.1 IMPLEMENTATION DETAILS

The input to the input processing phase neural network is a RGB image of size 84x84x3 and an instruction. The input image is processed with a CNN that has four convolution layers. The first layer convolves the image with 32 filters of 5x5 kernel size with stride 2, followed by another 32 filters of 5x5 kernel size with stride 2. This is then followed by 64 filters of 4x4 kernel size with stride 1 and finally by another 64 filters of 3x3 kernel size with stride 2. The input instruction is encoded through a GRU of size 16. The encoded instruction is then passed through a FC layer of size 64. The visual mode is then combined with textual mode through various multi-modal fusion mechanisms: a) simple concatenation, b) our attention mechanism and its variants, c) Gated-Attention unit (Hadamard product). In case of our attention mechanism, the multiple attention maps are concatenated and passed through two other convolutional layers (each having 64 filters of 3x3 kernel size with stride 1) before passing it to the LSTM layer. All our experiments are performed with A3C algorithm. Our policy learning phase has a LSTM layer of size 32, followed by fully connected layer of size 4 to estimate the policy function as well as fully connected layer of size 1 to predict value function. The LSTM layer enables the agent to have some memory of previous states. This is crucial as the agent receives egocentric view in which all the objects may not be present and hence need to remember the previously seen objects. The network parameters are shared for predicting both the policy function and the value function except the final fully connected layer. All the convolutional layers and FC layers have PReLU activations He et al. (2015). We observed during our experimentation the importance of not suppressing the negative gradients, as the same architecture in which all convolutional and FC layers have ReLU activations Nair & Hinton (2010) performed poorly on our evaluation scenarios (Refer section 5 for this comparison). The A3C algorithm was trained using Adam optimizer Kingma & Ba (2014) with an annealing learning rate schedule starting with 0.0001 and reducing by a fraction of 0.9 after every 10000 steps. For each experiment, we run 32 parallel threads and we use a discount factor of 0.99 for calculating expected rewards. The gradients of each of these worker threads are clipped in order to prevent overly-large parameter updates which can destabilize the policy. As described in Mnih et al. (2016) we use entropy regularization for improved exploration. Further, in order to reduce the variance of the policy gradient updates, we use the Generalized Advantage Estimator Schulman et al. (2015).

## 7 RESULTS AND DISCUSSIONS

Table 1 depicts the performance of our attention mechanism ($A3C_{attn}$) as described in section 5.2 for various $n$ values on unseen scenarios. The *accuracy* and *mean reward* values are averaged over 100 episodes. We observe from the table that $n = 5$ achieves best *mean reward* of 0.95. We also found that when $A3C_{attn}$ ($n = 5$) is evaluated under *zero-shot* generalization settings, it obtains a *mean reward* of 0.8. For *zero-shot* evaluation, we measure the agent's success rate on 19 instructions that were held out during training and these instructions consist of new combinations of attribute-object pairs not seen before by the agent during train phase (refer section 9.5 for these instructions). The results have been updated in the table 1. Figure 11a shows the *mean reward* obtained by our $A3C_{attn}$ model for different $n$ values as the training progresses. We observe from the graph that $n = 5$ converges faster to higher reward values. Figure 11b shows the comparison of our best performing $A3C_{attn}$ model in which all convolutional and FC layers have PReLU activations against the case in which all convolutional and FC layers have ReLU activations. It is evident from the graph the performance gain obtained just by not suppressing negative gradients using PReLU and accentuates the importance of choosing appropriate activation function. Table 2 portrays the perfor-

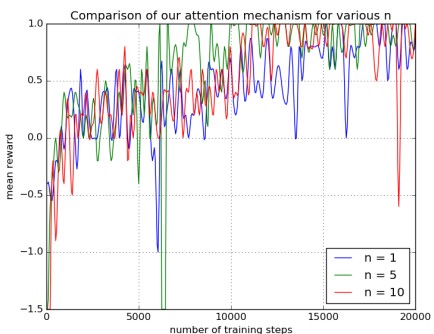

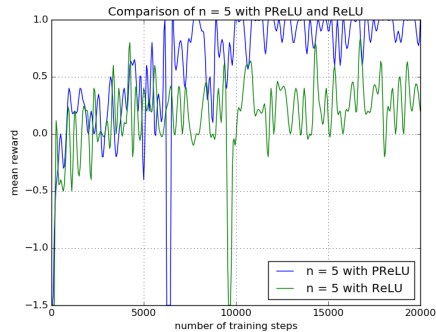

(a) Performance of $A3C_{attn}$ ($n = 1, 5, 10$).

(b) Performance of $A3C_{attn}$ ($n = 5$) with PReLU vs ReLU.

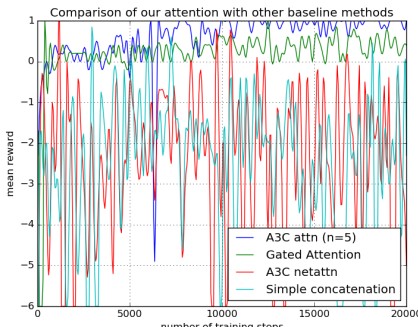

(c) Performance of $A3C_{attn}$ ($n = 5$) with other baseline methods.

Figure 5: Performance analysis of our attention mechanism and comparison with other baseline methods.

mance of $A3C_{attn}$ with $n = 5$ with a variant of our attention method ($A3C_{netattn}$) as well as with other previously proposed multimodal fusion mechanism methods ($A3C_{hadamard}$ and $A3C_{concat}$) as described in section 5.2. From figure 11c, it is apparent that our $A3C_{attn}$ with $n = 5$ converges faster and performs significantly better than other methods.

Table 1: The mean reward and accuracy of our model for different $n$ values on unseen scenario generalization and ZS instructions.

| Model | mean reward | mean reward (ZS) | accuracy |
|---|---|---|---|
| $A3C_{attn}$ ($n = 1$) | 0.87 | 0.73 | 0.86 |
| $A3C_{attn}$ ($n = 5$) | **0.95** | **0.8** | **0.92** |
| $A3C_{attn}$ ($n = 10$) | 0.94 | 0.75 | 0.88 |

Table 2: Comparison of our model with other baseline methods on unseen scenario generalization.

| Model | mean reward | accuracy |
|---|---|---|
| $A3C_{attn}$ ($n = 5$) | **0.95** | **0.92** |
| $A3C_{netattn}$ | -2.8 | 0.1 |
| $A3C_{hadamard}$ | 0.4 | 0.5 |
| $A3C_{concat}$ | -2.9 | 0.1 |

We also visualize the attention maps for our best performing $A3C_{attn}$ model. Figure 6 shows the visualization of attention maps for the case when the sentence is "Go to small red car." As shown in figures 6b, 6c and 6d, different attention maps focus on different regions in the environment and the

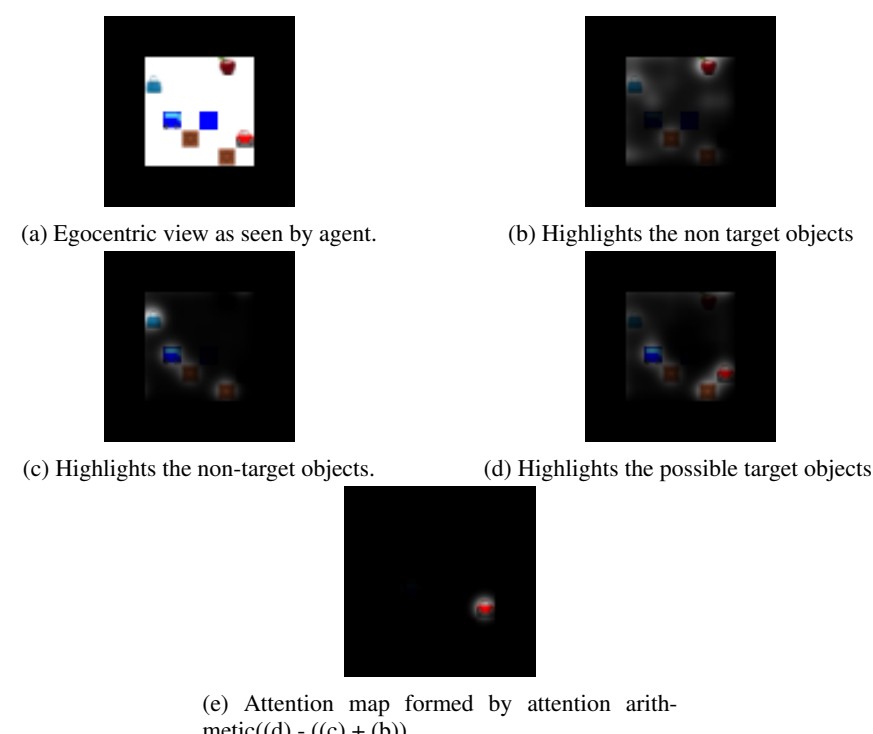

(a) Egocentric view as seen by agent.

(b) Highlights the non target objects

(c) Highlights the non-target objects.

(d) Highlights the possible target objects.

(e) Attention map formed by attention arithmetic((d) - ((c) + (b))

Figure 6: Visualization of attention maps for sentence: "Go to small red car."

agent uses a combination of these in order to learn the policy and successfully navigate to the target object while avoiding incorrect objects and obstacles. Attention map 6d highlights the possible target objects while attention maps 6b and 6c focus on non-target objects. Looking at the attention maps it seemed that the agent was subtracting the union of maps 6b and 6c from 6d to figure out the target(small red car). We did this arithmetic and generated the attention map show in figure 6e thus confirming our hypothesis. As mentioned previously in section 5.2, the above visualization also underscores the fact that through our attention mechanism we obtain a joint concise representation of both the textual and visual modalities which exemplifies consciousness Bengio (2017). Figure 7 provides another visualization of attention maps for the case when the sentence is "There are multiple tree. Go to smaller one." We found the same pattern to be consistent across many different examples whereby one attention map would focus on the possible target objects and other maps would highlight the non-target one. The visualization of attention weights clearly indicate that the extracted representation of instruction are semantically meaningful and grounded in various visual elements in the environment.

In order to understand the quality of instruction embedding learnt by the GRU, we do a two-dimensional Principal Component Analysis (PCA) projection of the original 16 dimension representation obtained when the input instruction is encoded through a GRU of size 16. As shown in figure 8a, *vector("Go to green apple") - vector("Go to apple")* is parallel to *vector("Go to green sofa") - vector("Go to sofa")*. Similarly *vector("Go to blue bag") - vector("Go to bag")* is parallel to *vector("Go to blue bus") - vector("Go to bus")*. We also observe a similar pattern with instructions that include size attribute in addition to color attribute. From figure 8b, we discern that *vector("Go to medium blue bag") - vector("Go to blue bag")* is parallel to *vector("Go to medium green tree") - vector("Go to green tree")*. Likewise, *vector("Go to small blue bag") - vector("Go to blue bag")* is parallel to *vector("Go to small green tree") - vector("Go to green tree")*. In figure 8c, we notice similar relationship with instructions that specify direction. These evidently highlight our model's ability to organize various concepts and learn the relationships between them implicitly.

Additionally, we also illustrate that these instruction embeddings follow vector arithmetic. In order to elucidate this, we obtain the embeddings for the instruction *"Go to small green tree"* as follows: *vector("Go to tree") + (vector("Go to green apple") - vector("Go to apple")) + vector("Go to*

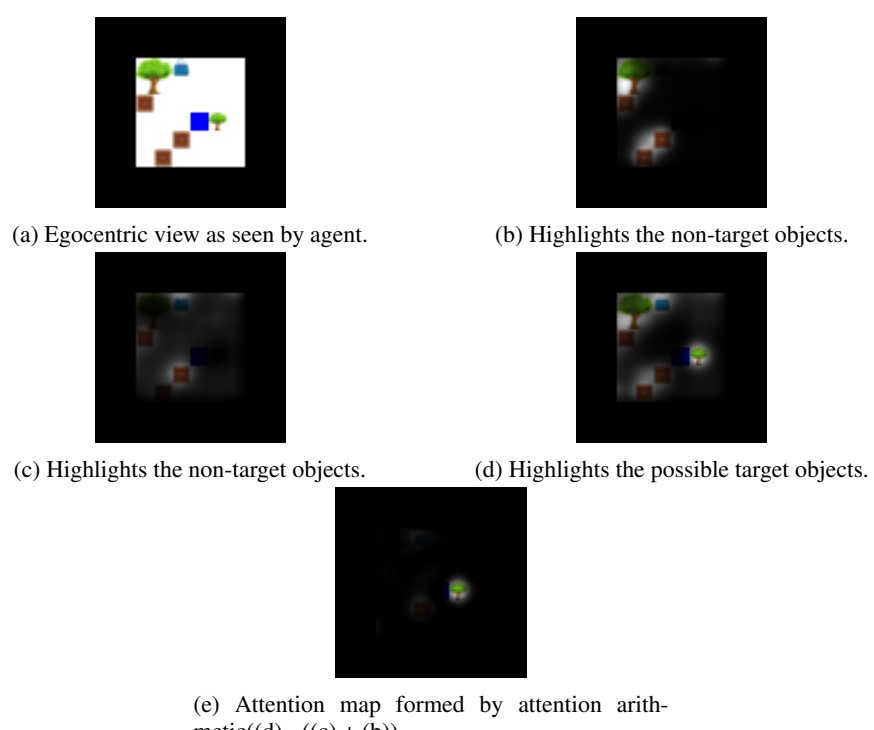

(a) Egocentric view as seen by agent.

(b) Highlights the non-target objects.

(c) Highlights the non-target objects.

(d) Highlights the possible target objects.

(e) Attention map formed by attention arithmetic((d) - ((c) + (b))

Figure 7: Visualization of attention maps for sentence: "There are multiple tree. Go to smaller one."

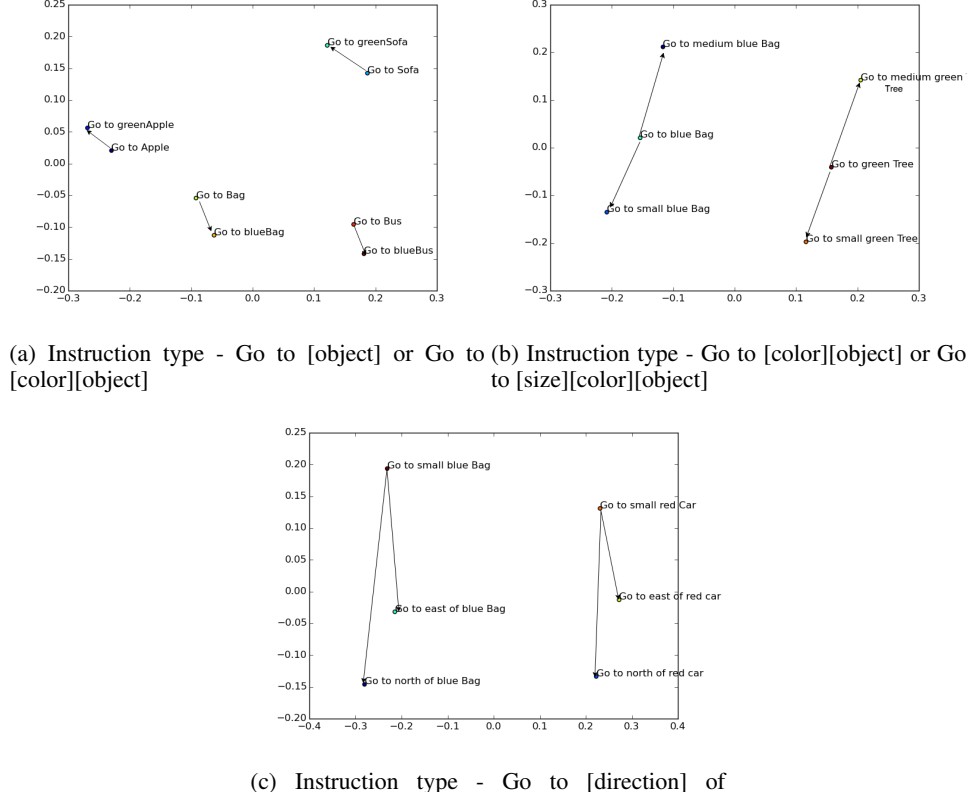

(a) Instruction type - Go to [object] or Go to [color][object]

(b) Instruction type - Go to [color][object] or Go to [size][color][object]

(c) Instruction type - Go to [direction] of [color][object] or Go to [size][color][object]

Figure 8: Two-dimensional PCA projection of instruction embedding learnt by the GRU.

*small red car") - vector("Go to red car").* Using this, the agent learns to navigate successfully to the small green tree as depicted in 9. Figure 9a is the complete state of the environment consisting of small green tree, medium green tree, small red car and medium red car. Figure 9b shows the initial egocentric view of the agent. Although the larger green tree is visible to the agent in its initial view and the smaller green tree is not visible, the agent learns to avoid the larger green tree and navigate correctly to the smaller green tree. Figure 9c shows an intermediate state in the agent's trajectory to the target object. In figure 9d, the agent is next to the target object and it finally reaches the target object in figure 9e. We found that we could do other interesting things with the embedding vectors. For example using the embedding calculated by *vector("There are multiple green Tree. Go to smaller one") - (vector("Go to small blue Bag") - vector("Go to blue Bag")) + (vector("Go to medium blue Bag") - vector("Go to blue Bag"))*, makes the agent go to medium green tree even when small green tree is present in the environment as shown in figure 10. The agent even responds to sentences such as "Go to small Apple" formed by *vector("Go to Apple") + vector("Go to small green Sofa") - vector("Go to green Sofa")* even though the agent has never seen sentences of the form *"Go to [size][object]"*.

### 7.1 ATTEMPT AT LANGUAGE TRANSLATION AS A BY-PRODUCT OF GROUNDING

Language grounding eventually means capturing the essence of words which consequently should enable an agent to translate the same concept across different languages. For e.g if an agent knows the concept 'apple' in multiple languages represent the same object. Then given any word which represent the concept 'apple' in any of its known languages, the agent should be able to translate it other known languages. In order to examine if our agent is grounded well enough to do such translations, we train the agent by giving it instructions in both English and French on a subset of vocabulary. At test time, to assess the instruction embedding, we ask the agent to translate English instructions to French and vice-versa. We found that the agent manages to translate **85%** of the instructions.

To achieve the above task, we attach a decoder branch consisting of two separate decoders, one for each English and French, on top of the instruction embedding obtained through GRU. The task of the decoders during the training phase was to reconstruct the original instruction as it is. Based on whether the natural language instruction is English or French, reconstruction loss through the corresponding decoder is back-propagated to modify the weights of the GRU. However during the test phase the decoder corresponding to each language is switched. Therefore the English decoder is activated while giving French instructions and the French decoder is activated while giving English instructions. It is important to note that the translation was done in a completely unsupervised way, without explicitly giving parallel corpora to the agent.

Thus, all these experiments prove that the agent has learnt to correctly associate the words with their true sense or meaning entirely on its own. In all these experiments, the agent's trajectory as it navigates to the target object are stored in the form of GIFs and are available https://github.com/rl-lang-grounding/rl-lang-ground.

## 8 CONCLUSION AND FUTURE WORK

In the paper we presented an attention based simple architecture to achieve grounding of natural language sentences via reinforcement learning. We show that retaining just the representation obtained after multimodal fusion phase (i.e. multiple attention maps) and discarding the visual features helps the agent achieve its goals. We justify this claim by visualization of the attention maps which reveal that they contain sufficient information needed for the agent to find the optimal policy. Through vector arithmetic, we also show that the embeddings learnt by the agent indeed make sense. In order to encourage the research in this direction we have also open sourced our environment as well as the code and models developed.

Our environment is capable of supporting rich set of natural language instructions and it's highly flexible. As a future work, we would like to increase the complexity of both the types of sentences generated as well as the environment dynamics by introducing moving objects. We also plan to take forward our approach to a 3D environment to test how well does it extend there.

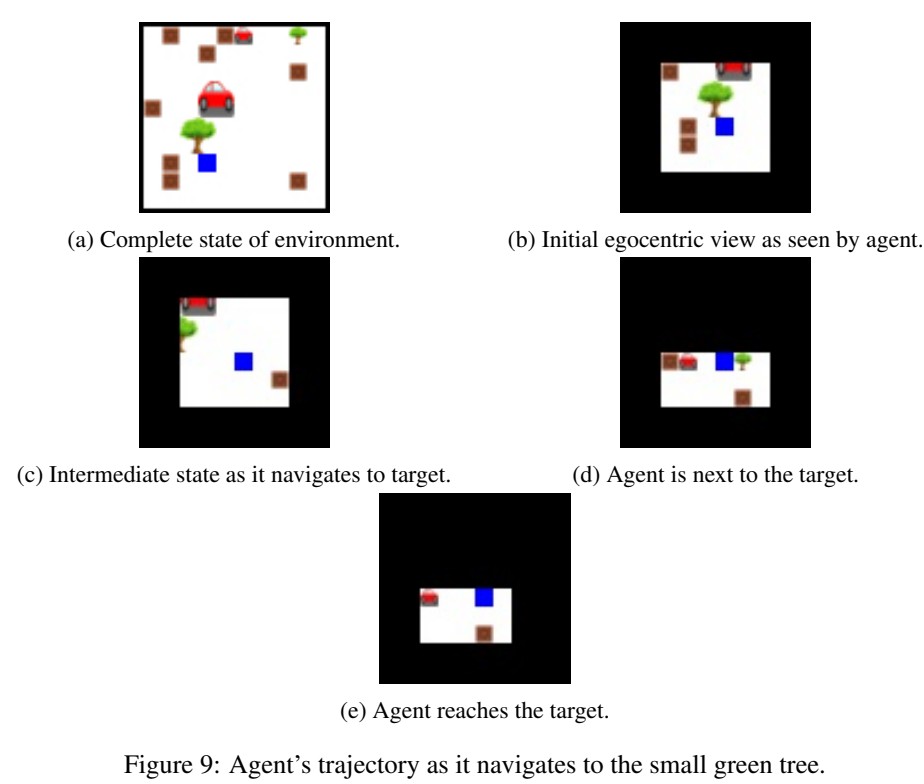

(a) Complete state of environment.

(b) Initial egocentric view as seen by agent.

(c) Intermediate state as it navigates to target.

(d) Agent is next to the target.

(e) Agent reaches the target.

Figure 9: Agent's trajectory as it navigates to the small green tree.

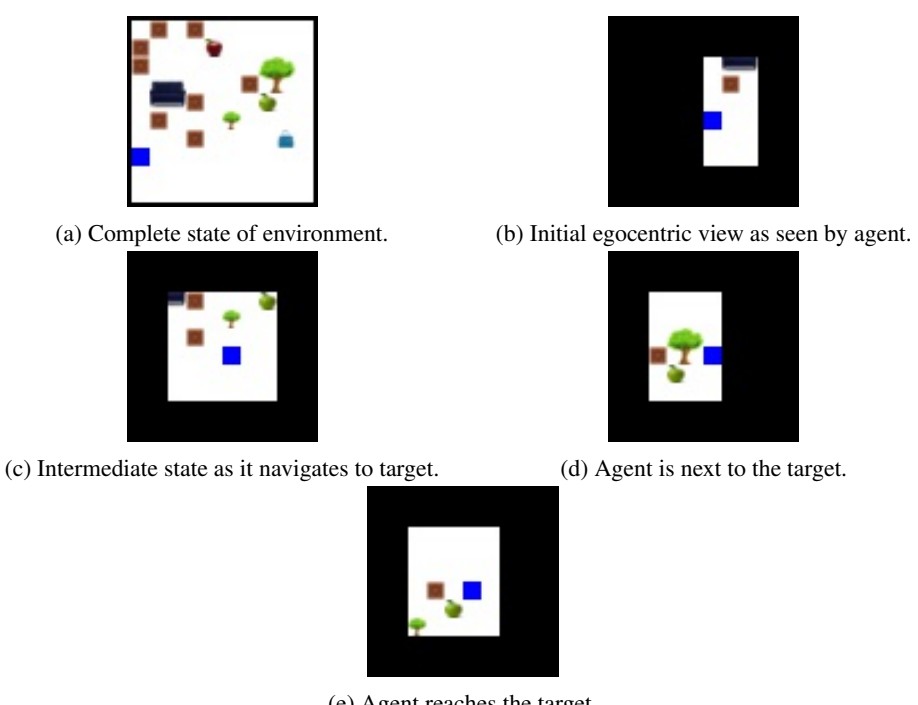

(a) Complete state of environment.

(b) Initial egocentric view as seen by agent.

(c) Intermediate state as it navigates to target.

(d) Agent is next to the target.

(e) Agent reaches the target.

Figure 10: Agent's trajectory as it navigates to the medium green tree.

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

# 9 APPENDIX

## 9.1 RESULTS ON 3D ENVIRONMENT

To test if our attention mechanism could scale to 3D environments, we applied our fusion mechanism to the Vizdoom based environment and compared our results with Chaplot et al. (2017) who have recently open sourced their code Chaplot. We replaced their fusion mechanism with our attention based fusion mechanism and the results for the easy, medium and also hard scenario are shown in figure 11. We have also compared our accuracy values with the results mentioned by the authors in their paper, in table 3, whereby we show significant performance improvement during the test phase including for the zeros shot instructions. The experiments are still running for the hard difficulty case but the results show that our approach converges much faster than the original fusion mechanism used by them. We see that our mechanism thus scales to both 2D as well as 3D environments outperforming other baselines in both the scenarios. The experiments were conducted on the same set of hardware thus ensuring a fair comparison between the plots.The plot shown here differs from the one shown in the paper( Chaplot et al. (2017)) because of a possible difference in hardwares used. We will update the plots as soon as all the experiments have been completed.

Table 3: The mean accuracy comparison between our atttention mechanism results and the results mentioned by Chaplot et al. (2017) in their paper.

| Model | Easy | Easy(ZS) | Medium | Medium(ZS) |
|---|---|---|---|---|
| Our | 1.00 | **0.96** | **0.92** | **0.9** |
| Original | 1.00 | 0.81 | 0.89 | 0.75 |

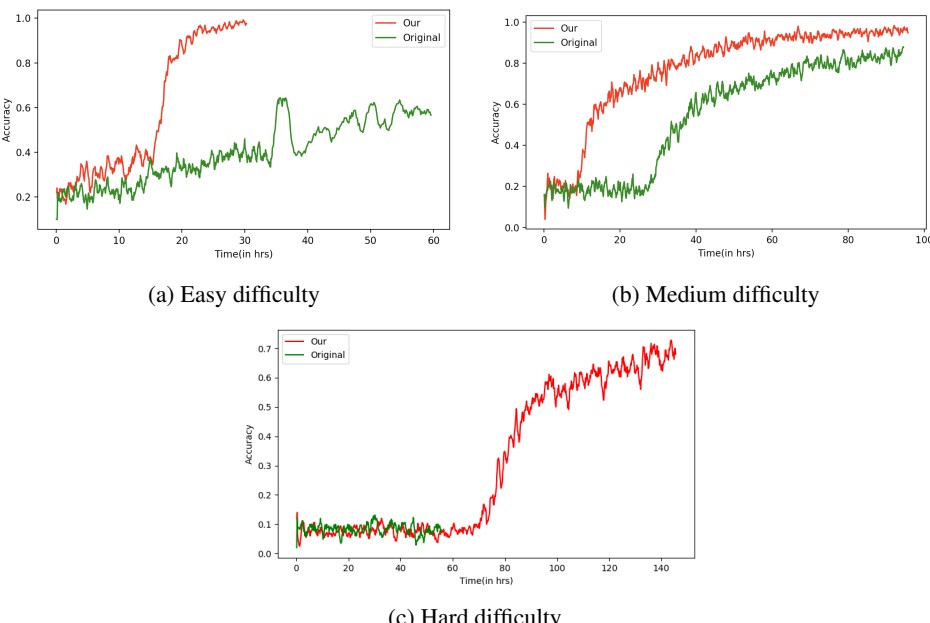

(a) Easy difficulty            (b) Medium difficulty

(c) Hard difficulty

Figure 11: Comparison of our attention based fusion mechanism with the fusion mechanism by Chaplot et al. (2017) for 3d environment.

## 9.2 LIST OF OBJECTS

Our 2D environment is completely customizable. The different objects, obstacles along with their corresponding attributes like color, size can be specified in a configurable specification file which is in JSON format. The images used to represent these objects are publicly available. The size attribute of each of the objects can be *small* (1x1), *medium* (2x2) or *large* (4x4). Figure 12 contains some of the different objects used for experiments in our environment.

## 9.3 LIST OF INSTRUCTIONS

The various natural language instructions that specify the characteristics of the target object based on which the agent learns to navigate in our 2D grid environment are listed below.

- *Go to [object]* - where [object] is one of the objects given in section 9.2, i.e. *apple*, *orange*, *sofa*, *car*, *chair*, *bus*, *bag* and *tree*.
- *Go to [color][object]* - where [color][object] pairs are present in *objects* list (section 9.2).
- *Go to [size][color][object]* - where [size] can be *small*, *medium* or *large* and [color][object] pairs are present in *objects* list (section 9.2).
- *Go to [direction] of [color][object]* - where [direction] can be *north*, *south*, *east* or *west* and [color][object] pairs are present in *objects* list (section 9.2).
- *Go to [top (or bottom)] [left (or right)] corner*.
- *There is a [color][object]. Go to it.* - where [color][object] pairs are present in *objects* list (section 9.2).
- *There is a [size][color][object]. Go to it.* - where [size] can be *small*, *medium* or *large* and [color][object] pairs are present in *objects* list (section 9.2).
- *There are multiple [color][object]. Go to smaller(or larger) one.* - where [color][object] pairs are present in *objects* list (section 9.2).
- *There is a [size1][color1][object1] and a [size2][color2][object2]. Go to former(or latter).* - where [size1], [size2] can be *small*, *medium* or *large* and [color1][object1], [color2][object2] pairs are present in *objects* list (section 9.2).

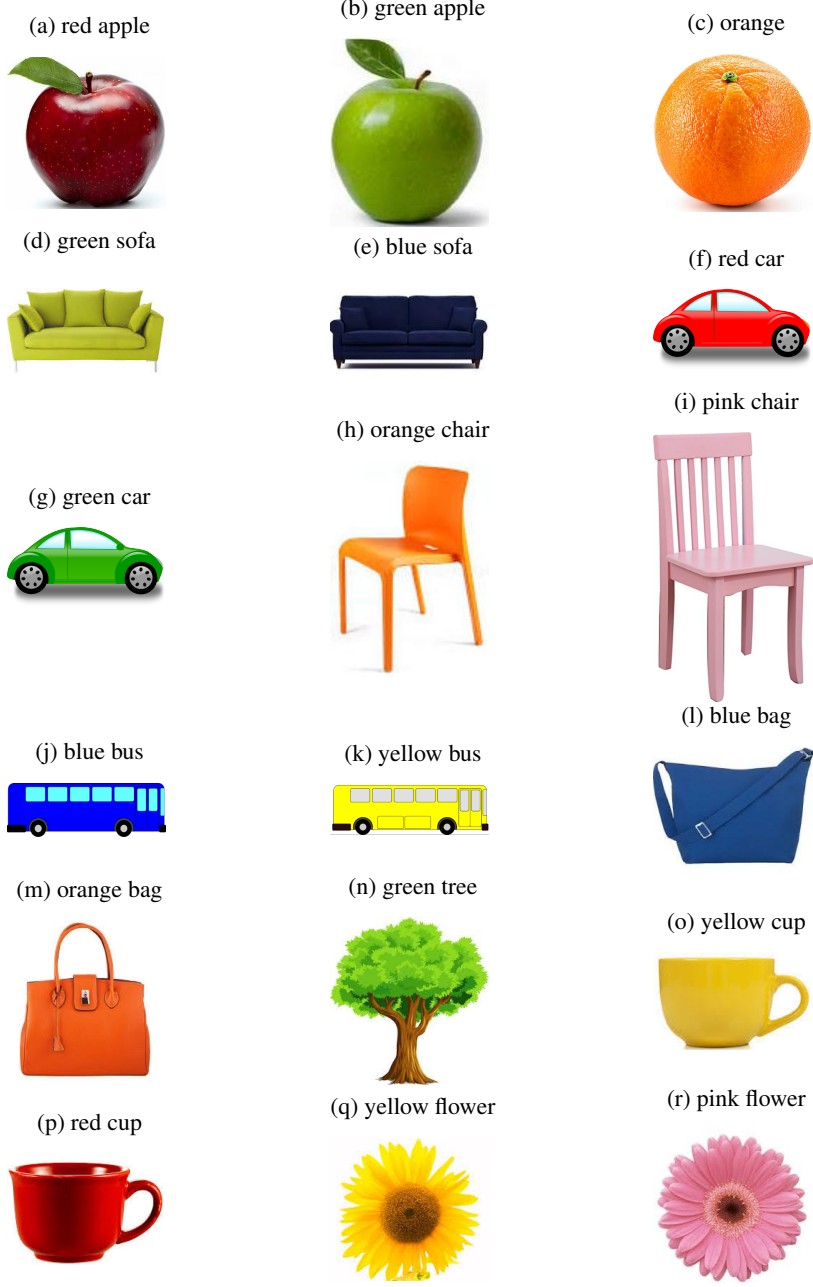

Figure 12: Some different objects used for experiments in our environment.

- *If [size1][color1][object1] is present then go to [size2][color2][object2] else go to [size3][color3][object3] - where [size1], [size2], [size3] can be small, medium or large and [color1][object1], [color2][object2], [color3][object3] pairs are present in objects list (section 9.2).*

## 9.4 VOCABULARY WORDS

The different unique words that can be present in instruction are: *vocab = {apple, orange, sofa, car, chair, bus, bag, tree, flower, cup, ball, knife, spoon, triangle, hexagon, grape, turtle, pig, fan, camel, carrot, television, banana, fridge, plane, top, pant, truck, mobile, shirt, washmachine, phone, laptop, go, navigate, move, to, the, of, there, is, and, are, a, it, red, green, orange, blue, pink, yellow, black, top, bottom, left, right, corner, north, south, west, east, small, medium, smaller, larger, one, multiple, former, latter, target, destination, your}*

## 9.5 ZERO SHOT GENERALIZATION INSTRUCTIONS

The list of unseen instructions used for evaluation under *zero-shot* generalization settings are: *instructions = {"Go to small red car", "Go to medium green apple", "Go to small orange", "Go to medium red cup", "There is a small red car. Go to it.", "There is a medium green apple. Go to it.", "There is a small orange. Go to it.", "There is a medium red cup. Go to it.", "Go to orange sofa", "Go to blue bus", "Go to yellow flower", "There is a orange sofa. Go to it.", "There is a blue bus. Go to it.", "There is a yellow flower. Go to it.", "Go to north of orange chair.", "Go to south of blue bus.", "Go to west of green tree.", "Go to east of blue sofa.", "Go to bag."}*

## 9.6 SAMPLE JSON SPECIFICATION

A sample JSON configuration file for an environment consisting of an agent, one object and one obstacle. More objects and obstacles can be added as desired.

```
1  {
2       "environment" :
3            {
4                    "Height" : 84,
5                    "Width" : 84,
6                    "Block_height" : 10,
7                    "Block_width" : 10,
8                    "Background" : [255,255,255],
9                    "Sizes" :{    "small" : 1,
10                                  "medium" : 2,
11                                  "large" : 3
12                           }
13           },
14
15      "agent" : [
16           {
17                    "img" : "agent.jpeg",
18                    "size" : "small",
19                    "step" : 1
20           }
21      ],
22
23      "objects" : [
24           {
25                    "id" : 1,
26                    "type" : "Apple",
27                    "color" : "green",
28                    "img" : "apple_crop.jpeg",
29                    "size" : "small",
30                    "hard" : 0,
31                    "reward" : 1
32           }
33      ],
34      "obstacles" :  [
35           {
36                    "id" : 2,
37                    "type" : "Walls",
38                    "color" : "orange",
39                    "img" : "brick_block.png",
40                    "size" : "small",
41                    "hard" : 1,
42                    "reward" : -1
43           }
44      ]
45  }
```

