# OpenReview forum: "Learning to navigate by distilling visual information and natural language instructions"
_ICLR.cc/2018/Conference — Reject_

### Official Review · AnonReviewer3 · 2017-11-22
**Toy experiments; Not much novelty; Missing references**

**Rating:** 4
**Confidence:** 4

**Review:**

Paper summary: The paper tackles the problem of navigation given an instruction. The paper proposes an approach to combine textual and visual information via an attention mechanism. The experiments have been performed on a 2D grid, where the agent has partial observation.

Paper Strengths:
- The proposed approach outperforms the baselines.
- Generalization to unseen combination of objects and attributes is interesting.

Paper Weaknesses:
This paper has the following issues so I vote for rejection: (1) The experiments have been performed on a toy environment, which is similar to the environments used in the 80's. There is no guarantee that the conclusions are valid for slightly more complex environments or real world. I highly recommend using environments such as AI2-THOR or SUNCG. (2) There is no quantitative result for the zero-shot experiments, which is one of the main claims of the paper. (3) The ideas of using instructions for navigation or using attention for combining visual and textual information have been around for a while. So there is not much novelty in the proposed method either. (4) References to attention papers that combine visual and textual modalities are missing.

More detailed comments:

- Ego-centric is not a correct word for describing the input. Typically, the perspective changes in ego-centric views, which does not happen in this environment.

- I do not agree that the attention maps focus on the right objects. Figures 6 and 7 show that the attention maps focus on all objects. The weights should be shown using a heatmap to see if the model is attending more to the right object.

- I cannot find any table for the zero-shot experiments. In the rebuttal, please point me to the results in case I am missing them.


Post Rebuttal:
I will keep the initial rating. The environment is too simplistic to draw any conclusion from. The authors mention other environments are unstable, but that is not a good excuse. There are various environments that are used by many users.

---

> ### Author Response · Authors · 2018-01-05
> **Reply to reviewer 3**
>
> We would like to thank you for your insightful comments. We have tried to address some of the concerns below.
>
> - The experiments have been performed on a toy experiment...
>
> Response - – We have updated the paper(section 2) highlighting how our environment is more complex than other 2d environments like the concurrent iclr submission ID 235 who focus on the task of language grounding. We tried performing the experiments over the AI2-THOR environment but found the environment to be unstable where the game would often stall down. We have further increased the complexity of our environment by working on a larger vocabulary set as well as on complex instructions.
>
> -There is no quantitative result for the zero-shot experiment
>
> Response - We have added the results in a tabular format in our latest revision(Table 1).
>
> - The ideas of using instructions for navigation or using attention for combining visual and textual information have been around for a while...
>
> Response - Our main aim was not to use instructions for navigation but rather to make an agent understand natural language for which we took the task of navigation.  Also we claim that our mechanism of fusing the visual and textual modalities is new because the same has not been used by researchers before. We had to come up with a new approach because the other approaches were not converging on our environment.
>
> -References to attention papers that combine visual and textual modalities are missing.
>
> Response - We have updated the references.
>
> - I do not agree that the attention maps focus on the right objects...
>
> Response - We have updated the plots corresponding to attention maps highlighting how the agent might be using arithmetic over the maps to figure out its policy.

---

> > ### Public Comment · (anonymous) · 2018-01-12
> > **Unsubstantial claim**
> >
> > "We have updated the paper(section 2) highlighting how our environment is more complex than other 2d environments like the concurrent iclr submission ID 235 ..."
> >
> > This claim is unsubstantial. The author made this claim without providing concrete statistics for comparison. As far as I can tell, submission 235 has a vocabulary size of 186, totaling ~1.6 million different sentences. There are 115 different object classes with 3 instances for each class. In contrast, this submission only has around 30 object classes. And from Appendix 9.2, I can see that the total number of sentences is far less than 1 million. Overall, even though the authors try to highlight their differences with several concurrent submissions, the arguments seem not convincing.

---

> > > ### Author Response · Authors · 2018-01-20
> > > **Reply to anonymous**
> > >
> > > In comparison to  ICLR submission ID 235 we would  claim that our environment is comparatively harder specifically because of the below mentioned reasons.
> > >
> > > Increased grid size:
> > > Grid size in our case is 10x10 as compared to 7x7 used by them.
> > >
> > > Number of objects present concurrently in the environment:
> > > The number of objects in the map ranges from 3 to 6 as compared to 1 to 5 used by them. More number of distractor objects makes it difficult for the agent to comprehend the correct goal.
> > >
> > > Increased complexity of natural language instructions:
> > > We have two sentence instructions and sentence length varies from 3 to 18 as compared to 2 to 13 in theirs.
> > >
> > > Size attribute of objects:
> > > In addition to having different instances of the same object that differ in their color (for example : green apple, red apple), we also have a size attribute associated with every object which can be either small (1x1), medium (2x2) or large (3x3). For example, if the environment has small red apple, medium red apple and large red apple and the sentence is “There are multiple red apple. Go to larger one.”, the agent receives positive reward only on successfully reaching the 3x3 red apple. On the other hand, if the environment only has small red apple and medium red apple with the same instruction, the agent receives positive reward only on successfully reaching the 2x2 red apple.
> > >
> > > We would like to highlight that our multimodal attention mechanism converged out of the box without any additional effort when we increased the vocabulary size from 40 (at the time of ICLR submission) to 72 (at the end of rebuttal period) and the code for the same has been updated in our Github repository. The number of objects can be further increased by picking up images of different objects from Google images or any other publicly available image source (on which we are currently working on).
> > > Also in case of 3d environments we provide experimental verification that our model works in 3d unlike submission 235.
> > >
> > > Our contribution in this work should also be seen from the implementation perspective as our model trains on a GPU, our environment is thread compatible (unlike the XWORLD2D used by them, source : https://github.com/PaddlePaddle/XWorld)

---

> ### Author Response · Authors · 2018-01-20
> **Results on 3D environment**
>
> We have updated the paper with results on vizdoom based 3D environment comparing our approach with other baseline in the appendix section. We show that our method converges much faster compared to the  baseline.

---

> ### Author Response · Authors · 2018-01-24
> **Additional improvements in 3D**
>
> We have added more results showing significant improvement over baseline both in terms of accuracy and speed of convergence.
> Also added preliminary results of hard scenario which looks promising ...

---

### Official Review · AnonReviewer2 · 2017-11-27
**Interesting Problem, but Limited Novelty and Flawed Evaluation**

**Rating:** 4
**Confidence:** 5

**Review:**

Interesting Problem, but Limited Novelty and Flawed Evaluation


The paper considers the problem of following natural language instructions given an first-person view of an a priori unknown environment. The paper proposes a neural architecture that employs an RNN to encode the language input and a CNN to encode the visual input. The two modalities are fused and fed to an RNN policy network. The method is evaluated on a new dataset consisting of short, simple instructions conveyed in simple environments.

The problem of following free-form navigation instructions is interesting and has achieved a fair bit of attention, previously with "traditional" structured approaches (rule-based and learned) and more recently with neural models. Unlike most existing work, this paper reasons over the raw visual input (vs., a pre-processed representation such as a bag-of-words model). HoA notable exception is the work of Chaplot et al. 2017, which addresses the same problem with a nearly identical architecture (see below). Overall, this paper constitutes a reasonable first-pass at this problem, but there is significant room for improvement to address issues related to the stated contributions and flawed evaluations.

The paper makes several claims regarding the novelty and expressiveness of the model and the contributions of the paper that are either invalid or not justified by the experimental results. As noted, a neural approach to instruction following is not new (see Mei et al. 2016) nor is a multimodal fusion architecture that incorporates raw images (see Chaplot et al.). The paper needs to make the contributions and novelty relative to existing methods clear (e.g., those stated in the intro are nearly identical to those of Mei et al. and Chaplot et al.). This includes discussion of the attention mechanism, for which the contributions and novelty are justified only by simple visualizations that are not very insightful. Related, the paper omits a large body of work in language understanding from the NLP and robotics domains, e.g., the work of Yoav Artzi, Thomas Howard, and Stefanie Tellex, among others (see below). While the approaches are different, it is important to describe this work in the context of these methods.


There are important shortcomings with the evaluation. First, one of the two scenarios involves testing on instructions from the training set. The test set should only include held-out environments and instructions, which the paper incorrectly refers to as the "zero-shot" scenario. This test set is very small, with only 19 instructions. Related, there is no mention of a validation set, and the discussion seems to suggest that hyperparameters were tuned on the test set. Further, the method is compared to incomplete implementations of existing baselines that admittedly don't attempt to replicate the baseline architectures. Consequently, it isn't clear what if anything can be concluded from the evaluation. There is a



Comments/Questions

* The action space does not include an explicit stop action. Instead, a run is considered to be finished either when the agent reaches the destination or a timeout is exceeded. This is clearly not valid in practice. The model should determine when to stop, as with existing approaches.

* The paper makes strong claims regarding the sophistication of the dataset that are unfounded. Despite the claims, the environment is rather small and the instructions almost trivially simple. For example, compare to the SAIL corpus that includes multi-sentence instructions with an average of 5 sentences/instruction (vs. 2); 37 words/instruction (vs. a manual cap of 9); and a total of 660 words (vs. 40); and three "large" virtual worlds (vs. 10x10 grids with 3-6 objects).

* While the paper makes several claims regarding novelty, the contributions over existing approaches are unclear. For example, Chaplot et al. 2017 propose a similar architecture that also fuses a CNN-based representation of raw visual input with an RNN encoding of language, the result of which is fed to a RNN policy network. What is novel with the proposed approach and what are the advantages? The paper makes an incomplete attempt to evaluate the proposed model against Chaplot et al., but without implementing their complete architecture, little can be inferred from the comparison.

* The paper claims that the fusion method realizes a *minimalistic* representation, but this statement is only justified by an experiment that involves the inclusion of the visual representation, but it isn't clear what we can conclude from this comparison (e.g., was there enough data to train this new representation?).

* It isn't clear that much can be concluded from the attention visualizations in Figs. 6 and 7, particularly regarding its contribution. Regarding Fig 6. the network attends to the target object (large apple), but not the smaller apple, which would be necessary to reason over their relative size. Further, the attention figure in Fig. 7(b) seems to foveate on both bags. In both cases, the distractor objects are very close to the true target, and one would expect the behavior to be similar irrespective of which one was being attended to.

* The conclusion states that the method is "highly flexible" and able to handle a "rich set of natural language instructions". Neither of these claims are justified by the discussion (please elaborate on what makes the method "highly flexible", presumably the end-to-end nature of the architecture) or the experimental results.

* The significance of randomly moving non-target objects that the agent encounters is unclear. What happens when the objects are not moved, as in real scenarios?

* A stated contribution is that the "textual representations are semantically meaningful" but the importance is not justified.

* Figure captions should appear below the figure, not at top.

* Figures and tables should appear as close to their first reference as possible (e.g., Table 1 is 6 pages away from its reference at the beginning of Section 7).


* Many citations should be enclosed in parentheses.



References:

* Artzi and Zettlemoyer, Weakly Supervised Learning of Semantic Parsers for Mapping Instructions to Actions, TACL 2013

* Howard, Tellex, and Roy, A Natural Language Planner Interface for Mobile Manipulators, ICRA 2014

* Chung, Propp, Walter, and Howard, On the performance of hierarchical distributed correspondence graphs for efficient symbol grounding of robot instructions, IROS 2015

* Paul, Arkin, Roy, and Howard, Efficient Grounding of Abstract Spatial Concepts for Natural Language Interaction with Robot Manipulators, RSS 2016

* Tellex, Kollar, Dickerson, Walter, Banerjee, Teller and Roy, Understanding natural language commands for robotic navigation and mobile manipulation, AAAI 2011

---

> ### Author Response · Authors · 2018-01-05
> **Reply to reviewer2**
>
> We would like to thank you for your insightful comments and suggestions regarding our work. We have tried to address the issues raised by you below. Our focus in this work was to achieve language grounding by make an agent understand natural language in a simulated environment, though further down the road we would want to take our work to real world. We have also updated the paper with additional information and modified language.
>
> 1) "The action space does not include an explicit stop action..."
>
> Response – Since our focus in this work is on language grounding, we did not need to include an explicit stop action. In our case, we are automatically getting a stop signal from our environment through the only positive reward signal. Such an approach is adopted by contemporary works on language grounding(Interactive Grounded Language Acquisition and Generalization in a 2D World, ICLR conference paper235).
>
>
> 2) "Despite the claims, the environment is rather smal......"
>
> Response – Our environment can be configured to increase the vocabulary, add many more instructions and increase the complexity of the environment accordingly. We have increased the complexity to a total of 72 and have also added several additional instructions (section 4) resulting in maximum 18 words in an instruction. We also like to point out that the environment is open source and over time, through public contributions, the size and complexity can be increased considerably.
>
> 3) "While the paper makes several claims regarding novelty, the contributions over existing approaches are unclear....."
>
> Response - Although our paper also uses a multi-modal fusion approach ,the exact fusion mechanism differs from other similar works. Chaplot et al. directly do a hadamard product between textual and visual embeddings, whereas we generate multiple textual embeddings by passing gru features to multiple fc layers and then use each of them to convolve with the visual features thus generating multiple attention maps. The idea behind generating multiple attention maps was to let each of them capture different environmental features such as which are the necessary objects and which are not. Furthermore, we found that the attention mechanism proposed by Chaplot et al. performed poorly on our environment (we have replicated the exact architecture now) and thus we had to come forward with a different fusion mechanism.
>
> 4) "The paper claims that the fusion method realizes a *minimalistic* representation..."
>
> Response - We say that this minimalistic representation is important because its leading to better representations of words that too with less memory overhead. We found that the attention maps, when concatenated with the visual features for finding the policy, did not lead to convergence.
>
> 5)" It isn't clear that much can be concluded from the attention visualizations...."
>
> Response – We have added attention maps in our latest revision in which objects are far apart. We show in the paper as to how the agent is using simple logic over attention maps to navigate to the correct target object.
>
> 6) "The conclusion states that the method is "highly flexible"...."
>
> Response - We would like to point out that we state that our environment is highly flexible(not the method). We say this because it’s very easy to add new objects in the environment through the json file. Moreover, one can also add new set of instructions via minimal changes in the code.
>
> 7) "The significance of randomly moving non-target objects that the agent encounters is unclear..."
>
> Response –The significance is twofold, one is that changing the position helped converge faster than fixing it. Secondly, we are helping the agent learn to avoid a non-target object through its visual appearance rather than memorizing its location. This property can be controlled via the value of the 'hard' attribute of each of the objects in the json.
>
> 8) " A stated contribution is that the "textual representations are semantically meaningful" but the importance is not justified..."
>
> Response - We say that the textual representations being semantically meaningful is important because it can make the agent respond to new combination of words that it had not seen before in training. If the agent knows what the words 'blue' and ’bag' actually mean then it can respond to 'Go to blue bag'  even without ever seeing it in training. We show those in last paragraph of section 7 wherein the agent responds to 'Go to [size][object]' type of sentences even without seeing them during training. We even attempted to do translation (section 7.1) to show the importance of language grounding.

---

> ### Author Response · Authors · 2018-01-20
> **Results on 3D environment.**
>
> We have updated the paper with results on vizdoom based 3D environment comparing our approach with other baseline in the appendix section. We show that our method converges much faster compared to the  baseline.

---

> ### Author Response · Authors · 2018-01-24
> **Additional improvements in 3D**
>
> We have added more results showing significant improvement over baseline both in terms of accuracy and speed of convergence.
> Also added preliminary results of hard scenario which looks promising ...

---

### Official Review · AnonReviewer1 · 2017-12-15
**While the paper aims to address an important and interesting problem, the novelty is limited and the experimental results are not convincing.**

**Rating:** 5
**Confidence:** 3

**Review:**

**Paper Summary**
The paper studies the problem of navigating to a target object in a 2D grid environment by following given natural language description as well as receiving visual information as raw pixels. The proposed architecture consists of a convoutional neural network encoding visual input,  gated recurrent unit encoding natural language descriptions, an attention mechanism fusing multimodal input, and a policy learning network. To verify the effectiveness of the proposed framework, a new environment is proposed. The environment is 2-D grid based and it consists of an agent, a list of objects with different attributes, and a list of obstacles. Agents perceive the environment throught raw pixels with a limited visible region, and they can perform actions to move in the environment to reach target objects.

The problem has been studied for a while and therefore it is not novel. The proposed framework is incremental. The proposed environment is trivial and therefore it is unclear if the proposed framework is able to scale up to a more complicated environment. The experiemental results do not support several claims stated in the paper. Overall, I would vote for rejection.


    - This paper solves the problem of navigating to the target object specified by language instruction in a 2D grid environment. It requires understanding of language, language grounding for visual features, and navigating to the target object while avoiding non-target objects. An attention mechanism is used to map a language instruction into a set of 1x1 convolutional filters which are intended to distinguish visual features described in the instruction from others. The experimental results show that the proposed method performs better than other methods.


    - This paper presents an end-to-end trainable model to navigate an agent through visual sources and natural language instructions. The model utilizes a proposed attention mechanism to draw correlation between the objects mentioned in the instructions with deep visual representations, without requiring any prior knowledge about these inputs. The experimental results demonstrate the effectiveness of the learnt textual representation and the zero-shot generalization capabilities to unseen scenarios.


**Paper Strengths**
- The paper proposes an interesting task which is a navigation task with language instructions. This is important yet relatively unxplored.
- The implementation details are included, including optimizers, learning rates with weight decayed, numbers of training epochs, the discount factor, etc.
- The attention mechanism used in the paper is reasonable and the learned language embedding clearly shows meaningful relationships between instructions.
- The learnt textual representation follows vector arithmetic, which enables the agent to perceive unseen instructions as a new combination of the attributes and perform zero-shot generalization.



**Paper Weaknesses**
- The problem of following natural language descriptions together with visual representations of environments is not completely novel. For example, Both the problem and the proposed method are similar to those already introduced in the Gated Attention method (Chaplot et al., 2017). Although the proposed method performs better than the prior work, the approach is incremental.

- The proposed environment is simple. The vocabulary size is 40 and the longest instruciton only consists of 9 words. Whether the proposed framework is able to deal with more complicated environments is not clear. The experimental results shown in Figure 5 is not convincing that the proposed method only took less than 20k iterations to perform almost perfectly. The proposed environment is small and simple compared to the related work. It would be better to test the proposed method in a similar scale with the existing 3D navigation environments (Chaplot et al., 2017 and Hermann et al., 2017).

- The novelty of the proposed framework is unclear. This work is not the first  one which proposes the multimodal fusion network incorporating a CNN achitecture dealing with visual information and a GRU architecture encoding language instructions. Also, the proposed attention mechanism is an obvious choice.

- The shown visualized attention maps are not enough to support the contribution of proposing the attention mechanism. It is difficult to tell whether the model learns to attend to correct objects. Also, the effectiveness of incorporating the attention mechanism is unclear.

- The paper claims that the proposed framework is flexible and is able to handle a rich set of natural language descriptions. However, the experiemental results are not enough to support the claim.

- The presentaiton of the experiment is not space efficient at all.

- The reference of the related papers which fuse multimodal data (vision and language) are missing.

- Comapred to 8 pages was the suggested page limit, 13 pages is a bit too long.

- Stating captions of figures above figures is not recommended.

- It would be better to show where each 1x1 filter for multimodal fusion attends on the input image.  Ideally, one filter should attend on the target object and others should attend on non-target objects. However, I wonder how RNN can generate filters to detect non-target objects given an instruction. Although Figure 6 and Figure 7 try to show insights about the proposed attention model, they don’t tell which kernel is in charge of which visual feature. Blurred attention maps in Figure 6 and 7 make it hard to interpret the behavior of the model.

- The graphs shown in the Figure 5 are hard to interpret because of their large variance. It would be better to smoothing curves, so that comparing methods clearly.

- For zero-shot generalization evaluation, there is no detail about the training steps and comparisons to other methods.

- A highly related paper (Hermann et al., 2017) is missing in the references.

- Since the instructions are simple, the model does not require attention mechanism on the textual sources. If the framework can take more complex language, might be worthwhile to try visual-text co-attention mechanism. Such demonstration will be more convincing.

- The attention maps of different attribute is not as clear as the paper stated. Why do we need several “non-target” objects highlight if one can learn to consolidate all of them?

- The interpretation of n in the paper is vague, the authors should also show qualitatively why n=5 is better than that of n=1,10. If the attention maps learnt are really focusing on different attributes, given more and more objects, shouldn’t n=10 have more information for the policy learning?

- The unseen scenario generalization should also include texture change on the grid environment and/or new attribute combinations on non-target objects to be more convincing.

- The contribution in the visual part is marginal.


** Preliminary Evaluation**
- The modality fusion technique which leads to the attention maps is an effective and seem to work well approach, however, the author should present more thorough ablated analysis. The overall architecture is elegant, but the capability of it to be extended to more complex environment is in doubt. The vector arithmetic of the learnt textual embedding is the key component to enable zero-shot generalization, while the effectiveness of this method is not convincing if more complex instructions such that it contains object-object relations or interactions are perceived by the agent.

---

> ### Author Response · Authors · 2018-01-05
> **Reply to reviewer1**
>
> We would like to thank you for your comments. We have tried to address the concerns raised by you below.
>
> - The problem of following natural language descriptions together with visual representations of environments is not completely novel...
>
> Response - We are not claiming that the problem is novel but the proposed method is simpler and trains faster compared to other approaches. Also, we found that all the mentioned approaches including Gated Attention method (Chaplot et al., 2017) didn’t work well in our simple 2d environment.
> Our simpler model consequently gives us insight into how the network is working with the help of internal attention maps. We have now shown in the paper that the network has most likely evolved a simple attention arithmetic logic to figure out which object it should go to.
> Another interesting result is the language translation capability the network has evolved without having trained on any parallel corpus.
>
> - The proposed environment is simple....
>
> Response - We have updated our paper with results over a larger vocabulary size of 72 with new objects and more complex sentences whereby the maximum words in an instruction are now 18.
>
> -The novelty of the proposed framework is unclear...
>
> Response - We would like to mention that for developing an end-to-end trainable multimodal deep-learning system the two basic components of
> 1. 	CNN based vision module
> 2. 	RNN based language module
> would be common across most papers. Approaches would however differ on fusion module which combines information from these modules to do the task.
> To the best of our knowledge, other researchers haven’t mentioned this attention based fusion approach in their works either as a proposal or as a baseline. In practice, this approach has worked better than all other methods for our task. It has also lead to more interesting results as mentioned earlier.
>
>
> - The shown visualized attention maps are ...
>
> Response - We have added new results which suggest that the network has evolved a simple attention arithmetic logic to figure out which object should it be interested in.
> The effectiveness of attention is clear because it leads to a smaller, faster and a better performing model.
>
>
> -The paper claims that the proposed framework is flexible ...
>
> Response  - We have added more complex sentences in our latest revision .
>
> - It would be better to show where each 1x1 filter for multimodal fusion attends on the input image...
>
> Response - We show each 1x1 filters output overlapped on the input image on figure 6,7. We think RNN is generating one filter which matches the target’s characteristics and other filters without those characteristics. In the paper the 6(d) and 7(d) filter focuses on possible targets and others on non targets. We have added  6(e) and 7(e) figures generated by plausible attention map arithmetic (for lack of a better term) which in our finding consistently seems to be the target object.
>
> - For zero-shot generalization evaluation, there is no detail about the training steps and comparisons to other methods.
>
> Response  - We have not added comparison to other methods as they performed very poorly on training set.
>
> - A highly related paper (Hermann et al., 2017) is missing in the references.
>
> Response - Referenced
>
> - Since the instructions are simple, the model does not require attention mechanism on the textual sources...
>
> Response - We will make this a part of our future work
>
> - The attention maps of different attribute is not as clear as the paper stated. Why do we need several “non-target” objects highlight if one can learn to consolidate all of them?
>
> Response - Yes, we have seen 2 or 3 attention maps works well and supports the argument of consolidated maps. But we need to complete all experiments before stating that in our paper.
>
> - The interpretation of n in the paper is vague, the authors should also show qualitatively why n=5 is better than that of n=1,10...
>
> Response - In the egocentric view there is a limited number of objects that can appear which may limit the number of attention maps needed. We need to do further experiments by increasing the ego-centric vision size and changing the number of attention maps.
>
> - The unseen scenario generalization should also include texture change on the grid environment and/or new attribute combinations on non-target objects to be more convincing.
>
> Response - Is a part of our future work

---

> ### Author Response · Authors · 2018-01-20
> **Results on 3D environment**
>
> We have updated the paper with results on vizdoom based 3D environment comparing our approach with other baseline in the appendix section. We show that our method converges much faster compared to the  baseline.

---

> ### Author Response · Authors · 2018-01-24
> **Additional improvements in 3d**
>
> We have added more results showing significant improvement over baseline both in terms of accuracy and speed of convergence.
> Also added preliminary results of hard scenario which looks promising ...

---

### Public Comment · (anonymous) · 2017-12-16
**Overall good paper, model learns policy described as target instructions but lacks evidences for attention map, zero shot generalization and at how many training iterations was results claimed**

****Summary****

The paper focuses on solving RL problem in which an agent learns to navigate to a particular target object in a given 2D environment. The environment is specified as an image, i.e. raw pixel values, and the target object is inferred from a text received by agent at the beginning of the episode. To learn the policy on both input image and the text, the paper proposes a multimodal framework which combines features from both the image and the text to generate attention maps based on which the agent then makes decisions which are sufficient to learn an effective policy.

The paper proposes a CNN architecture to extract features from the egocentric view of the environment. For extracting features from the textual description of the target, a sequence of GRUs is used, processing a concatenation of multiple one-hot encodings each corresponding to word in the input text. The embedding obtained from GRU is then passed through multiple fully connected layers. Finally, to combine the features of image and text, the paper claims that  best features are obtained by convolving both textual and visual features at the final layer. A3C architecture was used to learn the policy.

Our reviews are based on original paper submitted at ICLR 2018.

*****Strengths*****
An honest effort is made to ensure transparency of the work and the involved experiments. The training phase code released on GitHub matches with the explanation and parameters provided in the paper.

Both the model and the runtime environment were shown to support multithreading, as well as operation on a GPU.

The proposed model achieves excellent performance: a mean reward of 0.95, and an accuracy of 0.92 under unseen scenarios, which is higher than the comparable existing methods.

Through training, the model is able to generalize the semantics and the meaning of textual instructions as shown by the ability to apply vector arithmetic on the encoded vectors to generate other vectors. A similar combination of vectors were shown to result in similar model behaviour as intended.

The final policy learnt by the agent is able to reach the target objects, thus proving the model capability to capture the nuances of shape, size, location and type of objects.

*****Weaknesses*****
The code lacks comments and documentation. Some function are unused, and hard to reintegrate without original knowledge of the developers (e.g. generation of attention maps). Poor naming conventions (hard to interpret).

Model implementation lacks testing stage. How to force the model to test an environment with a particular instruction? How to input particular vector to the model and see its behaviour?

Zero-shot generalization lacks proof. Code/Paper does not talk about implementation details.

Model training stage lacks stopping criteria. Paper presents performance metrics without specifying the number of episodes the model was trained for.

Novelty claim is misleading. Similar work exists in the literature [1] [2].

*****Reproducibility Results*****
We achieved similar model performance results as reported in the original paper. However, the number of completed training episodes when presenting the findings was not specified, which makes it hard to reproduce the exact behaviour.

The overall trend for the reward does not converge at the claimed rate; we trained the model for 50K episodes and our observations indicate that the reward of the model does not converge at the rate depicted in Figure 5 of the original paper; it takes more than 10K iterations for reward to reach above 0.5 value.

The paper does not report the accuracy obtained in the zero-shot generalization scenarios. In general, the paper talks very little about how zero-shot generalization was ensured (it requires special treatment), and there is no code provided to apply/test this.

The paper does not mention the methodology used for creating the visualization of attention maps. The attention maps as mentioned in paper are of dimension 7x7x5, where 5 FC layers are used on embeddings generated from GRU, whereas, the size of an input image is 84x84x3. The mapping of the corresponding weights from attention layer to the original image is not discussed. The results of our improvised process to visualize attention maps did not match with the claimed results; the generated maps do not necessarily emphasize points of attention.

We were able to successfully confirm all claims about vector arithmetic, namely parallel vectors resulting from similar vector combinations, and generating vectors using other vectors.

*****References*****

[1] Haonan Yu, Haichao Zhang, and Wei Xu. A deep compositional framework for human-like language acquisition in virtual environment.

[2] Devendra Singh Chaplot, Kanthashree Mysore Sathyendra, Rama Kumar Pasumarthi, Dheeraj Rajagopal, and Ruslan Salakhutdinov. Gated-attention architectures for task-oriented language grounding.

---

> ### Author Response · Authors · 2018-01-05
> **Code for getting attention map updated along with zero shot instruction results**
>
> We would like to thank the people for taking an attempt to reproduce our results. We have now updated our github repository with the codes suggesting how to get the attention maps mentioned in our paper. We have also updated our paper with the results over zero shot instructions.

---

### Author Response · Authors · 2018-01-05
**Updated version with more complex environment, updated attention maps and translation**

We have updated our paper with the following improvements:-

1) Increased vocabulary size to 72 with new objects and new words present in instructions.

2) Complex instructions - The agent now responds to 'Go to former/latter' and 'IF.. ELSE..' types of sentences. The maximum number of words in an instruction is now 18 compared to 9 in our previous version.

3) We have updated attention maps for better visualization. We show how the agent seems to be using simple arithmetic over the attention maps to figure out the policy.

4)We show the importance of learning good grounding by attempting to do translation from English to French and vice-versa.

5) The paper has been updated to include more references and highlighting how our approach is different than the concurrent submissions attempting to do language grounding.

---

### Author Response · Authors · 2018-01-20
**Updated paper with result on 3D**

We would like to mention that we have now evaluated our approach on modified VizDoom 3D environment used by Chaplot et al. and have observed encouraging results. To our knowledge this is the only multimodal fusion method that works both on 2d and 3d environments. We also observed a minimum of 2x speed up in convergence compared to Chaplot et al.  We have added these results in the appendix section of our paper and would keep on updating them as and when we get them.

---

> ### Author Response · Authors · 2018-01-24
> **Additional improvements in 3d**
>
> We have added more results showing significant improvement over baseline both in terms of accuracy and speed of convergence.
> Also added preliminary results of hard scenario which looks promising ...

---

### Decision · Program_Chairs · 2018-01-29
**ICLR 2018 Conference Acceptance Decision**

**Decision:**

Reject

**Comment:**

This paper was reviewed by 3 expert reviews and received largely negative reviews, with concerns about the toy-ish nature of the 2D environments and limited novelty.

Since ICLR18 received multiple papers on similar topics, we took additional measures to ensure that papers were similar papers were judged under the same criteria. Specifically, we asked reviewers of (a) this paper and (b) of a concurrent submission that also studies language grounding in 2D environments to provide opinions on (b) and (a) respectively. Unfortunately, while they may be on similar topic and both working on 2D environments, we received unanimous feedback that (b) was much higher quality ("comparison with multiple baselines, better literature review, no bold claims about visual attention, etc). We realize this may be disappointing but we encourage the authors to incorporate reviewer feedback to make their manuscript stronger.